# Integration of two RAB5 groups during endosomal transport in plants

Emi Ito[1,2]*, Kazuo Ebine[3,4], Seung-won Choi[2], Sakura Ichinose[2], Tomohiro Uemura[1], Akihiko Nakano[1,5], Takashi Ueda[3,4,6]*

[1]Department of Biological Sciences, Graduate School of Science, The University of Tokyo, Tokyo, Japan; [2]Department of Natural Sciences, International Christian University, Tokyo, Japan; [3]Division of Cellular Dynamics, National Institute for Basic Biology, Okazaki, Japan; [4]Department of Basic Biology, SOKENDAI, Okazaki, Japan; [5]Live Cell Super-Resolution Imaging Research Team, RIKEN Center for Advanced Photonics, Wako, Japan; [6]Japan Science and Technology Agency, PRESTO, Saitama, Japan

**Abstract** RAB5 is a key regulator of endosomal functions in eukaryotic cells. Plants possess two different RAB5 groups, canonical and plant-unique types, which act via unknown counteracting mechanisms. Here, we identified an effector molecule of the plant-unique RAB5 in *Arabidopsis thaliana*, ARA6, which we designated PLANT-UNIQUE RAB5 EFFECTOR 2 (PUF2). Preferential colocalization with canonical RAB5 on endosomes and genetic interaction analysis indicated that PUF2 coordinates vacuolar transport with canonical RAB5, although PUF2 was identified as an effector of ARA6. Competitive binding of PUF2 with GTP-bound ARA6 and GDP-bound canonical RAB5, together interacting with the shared activating factor VPS9a, showed that ARA6 negatively regulates canonical RAB5-mediated vacuolar transport by titrating PUF2 and VPS9a. These results suggest a unique and unprecedented function for a RAB effector involving the integration of two RAB groups to orchestrate endosomal trafficking in plant cells.

DOI: https://doi.org/10.7554/eLife.34064.001

*For correspondence:
itoemi@icu.ac.jp (EI);
tueda@nibb.ac.jp (TU)

**Competing interests:** The authors declare that no competing interests exist.

## Introduction

Eukaryotic cells contain various single-membrane-bound organelles, each of which possesses distinctive constituents and functions. For an organelle to maintain a specific identity and function, the protein and lipid content must be strictly regulated; however, organelles are actively interconnected, exchanging various substances to fulfill complex and diverse cellular activities. The inter-organelle trafficking system mediated by vesicular and/or tubular trafficking intermediates plays an integral role in ensuring proper organelle function. A single round of membrane trafficking involves several sequential steps: budding of vesicles/tubules from a donor organelle, delivery of the transport vesicles, and tethering and fusion of the vesicles with the target membrane. RAB GTPase is an evolutionarily conserved key regulator of the targeting/tethering step and is also responsible for other cellular activities, such as organelle movement and inter- and intracellular signaling (*Saito and Ueda, 2009*; *Stenmark, 2009*). These functions of RAB GTPases are fulfilled through nucleotide state-dependent binding with specific sets of proteins, collectively called effector proteins. To ensure the organized and specialized functions of each RAB GTPase, interactions with effectors should be temporally, spatially, and combinatorially regulated for each trafficking event (*Grosshans et al., 2006*; *Zerial and McBride, 2001*).

The early endosome acts as a communication platform between the intracellular environment and the cell surface and/or extracellular space. RAB5 is a key regulator of a wide spectrum of early endosomal functions in animal cells, including homotypic fusion between early endosomes, endosomal

**eLife digest** Living cells often contain compartments that pass proteins, fats and other biological molecules to one another via a process called membrane trafficking. Endosomes are one of the key platforms of membrane trafficking. These structures accumulate molecules from the outside of the cell, sort them, and then redirect them back to the cell surface or send them to other compartments within the cell where they can be broken down. Proteins known as RAB5s regulate many of the activities of endosomes. Some are found in a wide range of organisms, including animals, fungi, and plants, and are referred to as the "canonical" RAB5 group. Another group of RAB5 proteins are unique to land plants and some green algae.

The existence of two RAB5 groups (i.e. canonical and plant-unique) is a distinctive feature of plant cells. In 2011, researchers showed that a plant-unique RAB5 could interfere with and counteract the activities of a canonical RAB5. However, it remained ambiguous how these proteins could do this.

To resolve this question, Ito et al. – who include several researchers from the 2011 study – set out to find proteins that interact with a plant-unique RAB5 from *Arabidopsis thaliana*. The experiments identified one partner of a plant-unique RAB5, which was named PUF2. Unexpectedly, further experiments revealed that PUF2 also regulates canonical RAB5. PUF2 was found on the surface of the endosome together with RAB5s and a protein that activates RAB5s. Notably, PUF2 also interacted with the activating factor and the inactive form of canonical RAB5.

Based on these findings, Ito et al. propose that PUF2 acts as a landmark to bring inactive canonical RAB5 close to its activating factor, which helps to activate canonical RAB5. They suggest that the plant-unique RAB5 also competitively binds to the landmark and blocks the canonical RAB5.

Membrane trafficking is a universal system for all living organisms, yet the system seems to be customized among different organisms. These new findings provide further evidence that land plants have evolved a unique mechanism to regulate the activities of their endosomes. The next step is to reconstruct how this system evolved and unravel its relevance to the evolution of plant-specific traits.

DOI: https://doi.org/10.7554/eLife.34064.002

motility, regulation of lipid metabolism at the endosomal membrane, and signal transduction via the early endosome (*Grosshans et al., 2006*; *Miaczynska et al., 2004*; *Zerial and McBride, 2001*). RAB5 is also conserved in a broad range of eukaryotic systems, suggesting its ancient origin in the evolution of eukaryotes (*Brighouse et al., 2010*; *Dacks and Field, 2007*; *Elias, 2010*; *Pereira-Leal and Seabra, 2001*).

Endocytic trafficking pathways demonstrate surprising diversity among lineages, representing uniquely acquired endosomal and endocytic functions in each lineage. The endocytic pathway in plant cells provides a striking example: internalized proteins are initially delivered to the *trans*-Golgi network (*Dettmer et al., 2006*), some parts of which mature into multivesiculated late endosomes that acquire RAB5 (*Scheuring et al., 2011*). Conversely, in animal cells, sequestered proteins are initially delivered to RAB5-positive early endosomes, which mature into RAB7-positive late endosomes, mediated by a RAB5 effector complex comprising SAND1/Mon1 and CCZ1 (*Kinchen and Ravichandran, 2010*). This complex, which is also conserved in plants, distinctly regulates endosomal transport from animal and yeast systems (*Ebine et al., 2014*; *Singh et al., 2014*; *Takemoto et al., 2018*). Plants are equipped with multiple vacuolar trafficking pathways that involve RAB5 and RAB7 in unique ways (*Bottanelli et al., 2012*; *Ebine et al., 2014*; *Singh et al., 2014*), further underpinning the diversification of endosomal trafficking systems between plant and animal systems. However, the mechanisms underlying diversified endosomal trafficking systems involving evolutionarily conserved RAB5 remain unresolved.

The presence of the plant-unique RAB5 group, ARA6/RABF1 group, is a remarkable feature of plant endosomal trafficking. *Arabidopsis thaliana* has three RAB5 members: two canonical RAB5 members, ARA7 (aka RABF2b) and RHA1 (aka RABF2a), which act in vacuolar and endocytic transport (*Dhonukshe et al., 2006*; *Ebine et al., 2011*; *Kotzer et al., 2004*; *Sohn et al., 2003*), and plant-unique ARA6 (aka RABF1) (*Ueda et al., 2001*). In lieu of cysteine residues that are

isoprenylated at the C-terminus, which are essential for membrane binding and functions of canonical RAB GTPases (*Seabra, 1998*), ARA6 harbors an extra stretch in the N-terminus, where this protein is N-myristoylated and palmitoylated to target a distinct subpopulation of endosomes from canonical RAB5 with substantial overlap (*Haas et al., 2007*; *Ueda et al., 2001*). ARA6 promotes formation of the SNARE complex, which contains plant-specific R-SNARE VAMP727 at the plasma membrane (*Ebine et al., 2011*), and an endosomal function has also been described; overexpression of the nucleotide-free mutant form of ARA6 results in impaired vacuolar trafficking (*Bolte et al., 2004*; *Bottanelli et al., 2012*).

Despite their distinct functions, ARA6 and canonical RAB5 share the common activating factor, VPS9a (*Goh et al., 2007*). Intriguingly, loss-of-function mutations of ARA6 and canonical RAB5 confer counteracting effects in a *vps9a* mutant, further highlighting the distinct functions of these two plant RAB5 groups (*Ebine et al., 2011*), although the molecular mechanism integrating these two groups in endosomal trafficking remains unexplored.

In the present study, we identified and characterized the first effector molecule of ARA6, PLANT-UNIQUE RAB5 EFFECTOR 2 (PUF2). Based on these results, PUF2 is a key integrator of the two RAB5 groups in the unique endosomal trafficking system of plants.

## Results

### PUF2 interacts with plant-specific RAB5

Consistent with the notion that endosomal trafficking pathways in animals and plants have substantially diverged, close homologs of well-characterized RAB5 effectors, such as EEA1 and Rabaptin-5, do not exist in plants. To identify effector molecules of ARA6, we performed yeast two-hybrid screening using a GTP-fixed mutant of ARA6 (ARA6$^{Q93L}$) as bait. After screening 4.57 × 10$^5$ independent clones, we obtained a candidate clone encoding the C-terminal region of At1g24560 (*Figure 1A*). *At1g24560* encodes a 678-amino-acid protein of unknown function, without clear homologs in animals or yeasts. We designated this protein PUF2. Bacterially expressed and purified full-length PUF2 also bound to GST-tagged ARA6$^{Q93L}$ but not to GST-ARA6$^{S47N}$ or -ARA7$^{Q69L}$ *in vitro* (*Figure 1B*). Surprisingly, PUF2 also interacted with ARA7$^{S24N}$ (*Figure 1B*). Based on a co-immunoprecipitation analysis using a lysate prepared from a transgenic plant expressing PUF2-GFP and an anti-GFP antibody, PUF2-GFP and ARA6 form a complex *in planta*, whereas canonical RAB5 members ARA7 and RHA1 were not co-precipitated with PUF2-GFP under these experimental conditions (*Figure 1C*, left panels). However, when a crosslinker (dithiobis succinimidyl propionate; DSP) was added to the reaction, canonical RAB5 was also co-precipitated (*Figure 1C*, right panels). Thus, PUF2 interacts with active GTP-bound ARA6, and PUF2 also weakly and/or transiently forms a complex with GDP-bound canonical RAB5.

Four coiled-coil domains were predicted in the PUF2 protein using a simple modular architecture research tool (SMART, http://smart.embl.de/) (*Letunic et al., 2015*; *Schultz et al., 1998*), although PUF2 contained no known functional domain. We subsequently examined whether these coiled-coil regions are responsible for the interaction with ARA6. Truncated PUF2 containing only the fourth coiled-coil region was isolated in the yeast two-hybrid screening. Consistently, truncated PUF2 containing the other coiled-coil region did not interact with ARA6 (*Figure 1D*). Furthermore, full-length PUF2 did not interact with ARA6, which may indicate that these regions negatively regulate the interaction between ARA6 and PUF2. We did not detect interactions between any PUF2 constructs and ARA7 (*Figure 1—figure supplement 1*).

### PUF2 preferentially colocalizes with canonical RAB5 on endosomes

To investigate the subcellular localization of PUF2, we observed transgenic plants expressing GFP-tagged PUF2 under the regulation of its own regulatory elements (promoter, introns, and terminator), thus retaining the authentic function of PUF2 as described below. As shown in *Figure 2A*, PUF2 localized to punctate organelles in the cytoplasm, which dilated with the application of wortmannin (Wm; a phosphatidylinositol-3 and -4 kinase inhibitor) and aggregated into so-called BFA bodies with brefeldin A (BFA; an ARF GEF inhibitor) treatment. These drug responses were similar to those of multivesicular endosomes bearing ARA6 and/or ARA7 (*Ebine et al., 2011*; *Grebe et al., 2003*;

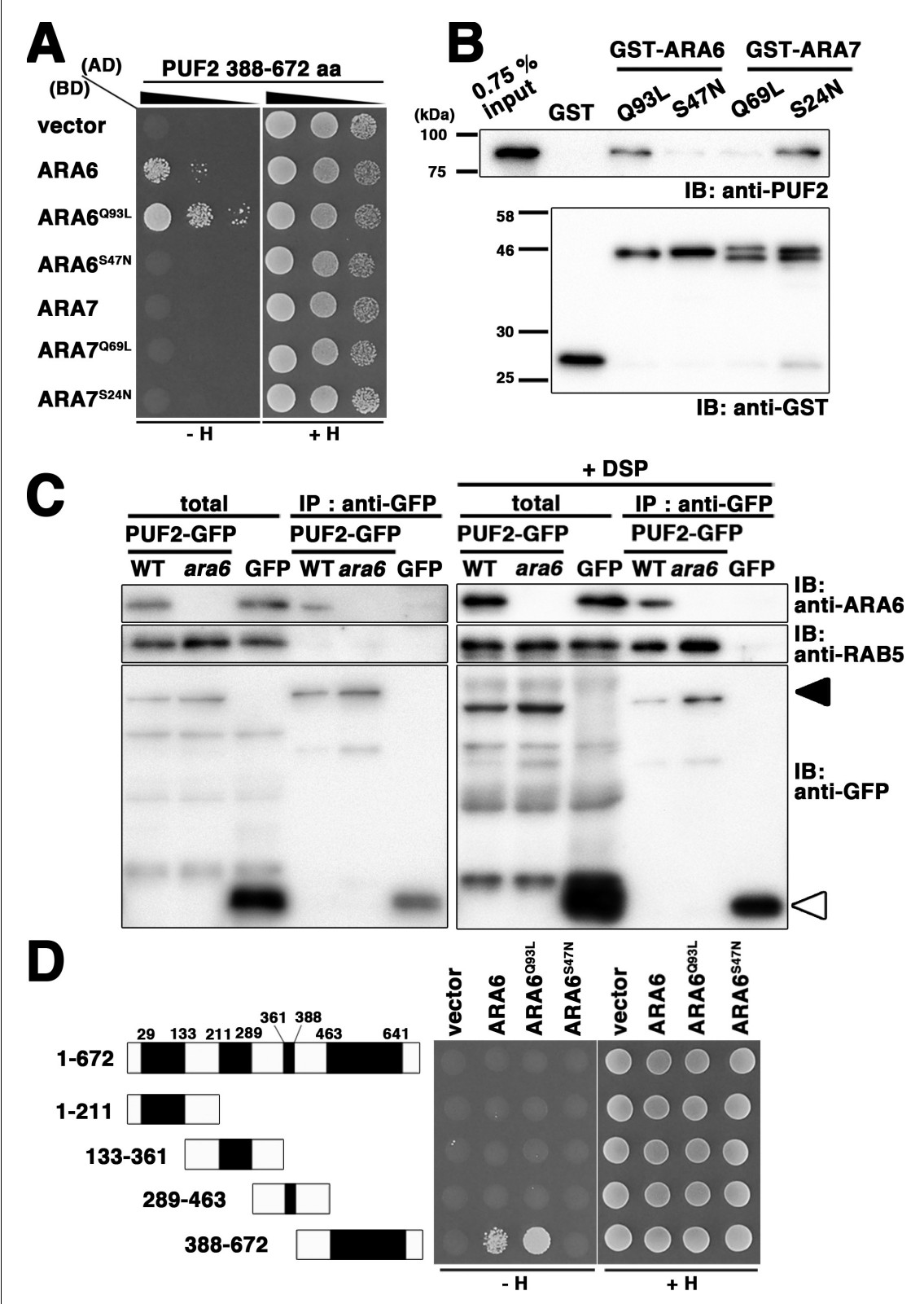

**Figure 1.** PLANT-UNIQUE RAB5 EFFECTOR 2 (PUF2) isolated as an ARA6 effector. (**A**) Yeast two-hybrid interaction between the C-terminal region of PUF2 and RAB5s (ARA6 or ARA7). PUF2 388–672 aa was expressed as a fusion protein with a transcriptional activation domain (AD), and ARA6 and ARA7 with the indicated mutations were expressed as fusion proteins with DNA binding domains (BDs) in the yeast strain AH109. Interactions with the *HIS3* reporter gene were examined. Q to L mutations are GTP-freezing mutations, and S to N mutations are GDP-freezing mutations. (**B**) Interaction

*Figure 1 continued on next page*

*Figure 1 continued*

between PUF2 and RAB5 members detected in an *in vitro* binding assay. The full-length PUF2 protein was pulled down using GST-tagged RAB5 proteins, which were immobilized in GTP- or GDP-bound states. (C) Wild-type or *ara6-1* transgenic plants expressing PUF2-GFP (black arrowhead) or free GFP (white arrowhead) were subjected to immunoprecipitation analyses with or without a chemical crosslinker (DSP) using an anti-GFP antibody, followed by immunoblotting with the indicated antibodies. A mixture of anti-RHA1 and anti-ARA7 antibodies was used to detect canonical RAB5. (D) Yeast two-hybrid interaction between truncated PUF2 proteins containing different coiled-coil regions and ARA6. Black boxes indicate coiled-coil regions.

DOI: https://doi.org/10.7554/eLife.34064.003

The following figure supplement is available for figure 1:

**Figure supplement 1.** Related to *Figure 1*.
DOI: https://doi.org/10.7554/eLife.34064.004

*Ito et al., 2012*; *Jaillais et al., 2008*). The endosomal nature of the PUF2-positive compartments was also supported by their accessibility to an endocytic tracer FM4-64 (*Figure 2A*, arrowheads).

We next compared the subcellular localization of PUF2 and ARA6 or ARA7 in transgenic plants coexpressing fluorescently tagged proteins. PUF2 exhibited good colocalization with both RAB5

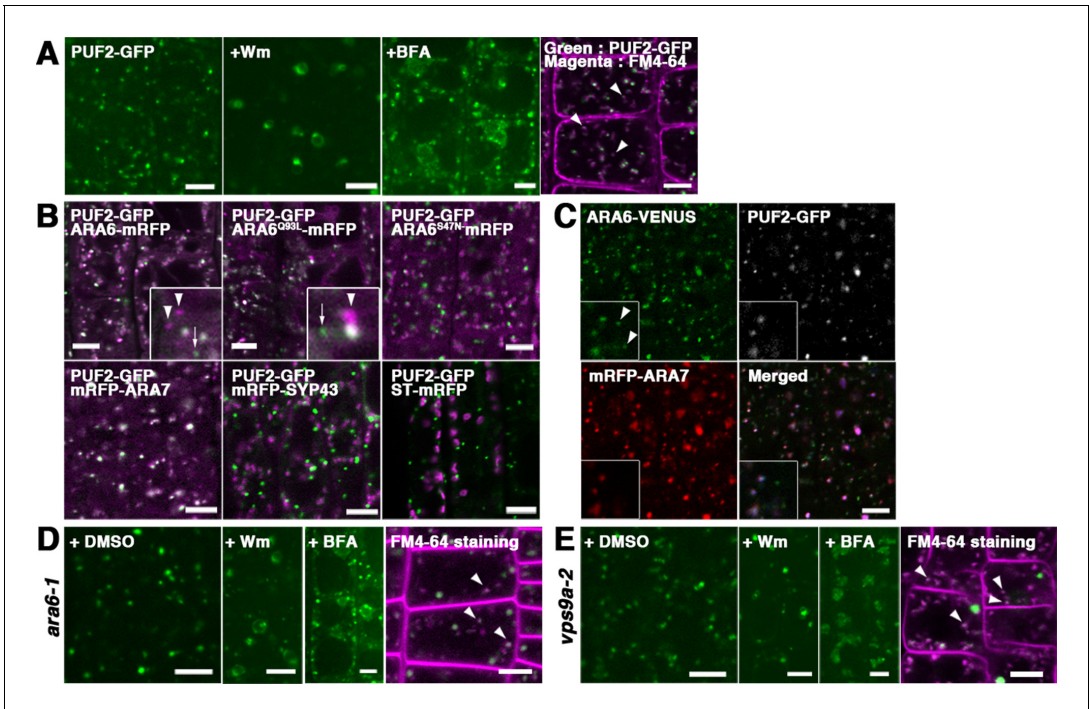

**Figure 2.** PUF2 preferentially colocalizes with ARA7. (A) Subcellular localization of PUF2-GFP in root epidermal cells (far-left panel). PUF2-localizing compartments were sensitive to the PI3K inhibitor Wm and an ARF GEF inhibitor, BFA. The PUF2-bearing compartments (green) were accessible to the endocytic tracer FM4-64 (magenta) (far-right panel). Bars = 5 μm. (B) Localization pattern of PUF2-GFP in relation to mRFP-tagged ARA6, ARA6$^{Q93L}$, ARA6$^{S47N}$, ARA7, SYP43 (*trans*-Golgi network marker), and ST (*trans*-Golgi marker). Puncta bearing only PUF2 and ARA6/ARA6$^{Q93L}$ are indicated with arrows and arrowheads, respectively. Bars = 5 μm. (C) Localization of PUF2-GFP (gray or blue), ARA6-Venus (green), and mRFP-ARA7 (red) in transgenic plants expressing all three tagged proteins. Arrowheads indicate endosomes bearing only ARA6. Bars = 5 μm. (D and E) Responses of PUF2-positive endocytic compartments to Wm or BFA in the *ara6-1* (D) and *vps9a-2* (E) mutants. PUF2-GFP (green) localized to endocytic compartments stained with FM4-64 (magenta) in these mutants (far-right panels). Bars = 5 μm.

DOI: https://doi.org/10.7554/eLife.34064.005

The following source data and figure supplements are available for figure 2:

**Figure supplement 1.** Related to *Figure 2*.
DOI: https://doi.org/10.7554/eLife.34064.006

**Figure supplement 1—source data 1.** Quantification of colocalization.
DOI: https://doi.org/10.7554/eLife.34064.007

members (*Figure 2B*). PUF2 also colocalized with GTP-fixed ARA6 but not with GDP-fixed ARA6. Interestingly, we occasionally observed PUF2-positive puncta without ARA6 (*Figure 2B*, arrows in upper panels), whereas PUF2 demonstrated almost complete colocalization with ARA7 (*Figure 2B*, lower left panel). Quantification of the colocalization confirmed our observation; specifically, 83.4 ± 4.2% (mean ±SD; n = 4 independent images, including 70 to 377 endosomes) of PUF2-GFP puncta overlapped with ARA6-mRFP, whereas 96.7 ± 1.7% (mean ±SD; n = 4 independent images, including 151 to 304 endosomes) of PUF2-GFP puncta colocalized with mRFP-ARA7 (*Figure 2—figure supplement 1*). Thus, PUF2 colocalized more fully with ARA7 than with ARA6 (*p*<0.05, Tukey's test). The colocalization ratio between ARA7 and PUF2 was comparable to that between GFP- and mRFP-tagged ARA6, indicating nearly perfect colocalization. Next, we generated transgenic plants coexpressing PUF2-GFP, ARA6-Venus, and mRFP-ARA7 and again observed endosomes bearing only ARA6 (*Figure 2C*, arrowheads), whereas PUF2 exhibited nearly perfect colocalization with ARA7 (*Figure 2—figure supplement 1*). PUF2 did not colocalize with the *trans*-Golgi network marker mRFP-SYP43 and the *trans*-Golgi cisternae marker ST-mRFP (*Boevink et al., 1998*; *Uemura et al., 2012*; *Uemura et al., 2004*; *Wee et al., 1998*); *Figure 2B*, lower middle and right panels).

## Endosomal localization of PUF2 does not require RAB5 activity

To determine whether the endosomal localization of PUF2 requires RAB5 activity, we observed the localization of PUF2-GFP expressed in mutants of ARA6 and the common activator of all RAB5 members in Arabidopsis, VPS9a (*Goh et al., 2007*). In *ara6-1* mutant plants, PUF2-GFP localized to punctate compartments, which were swollen and aggregated after treatment with Wm and BFA, respectively, and were labeled with FM4-64 as observed in wild-type plants (*Figure 2D*, arrowheads). PUF2-GFP also exhibited the same responses to drugs and FM4-64 accessibility in the *vps9a-2* mutant (*Figure 2E*). Thus, ARA6 and canonical RAB5 activities are not required for the recruitment of PUF2 onto endosomes.

## Cooperation between PUF2 and RAB5 is required for plant development

We next examined the effects of the loss of PUF2 function using a *puf2* mutant, in which PUF2 was not detected at either the mRNA or protein level (*Figure 3—figure supplement 1A–C*). The *puf2* mutant exhibited no macroscopic abnormalities under normal laboratory conditions (*Figure 3—figure supplement 1D*). Thus, we explored the potential genetic interactions between *PUF2* and the two *RAB5* groups. The *puf2 ara6-1* double mutant was indistinguishable from wild-type, *puf2*, and *ara6-1* plants. By contrast, the *puf2 rha1* double mutation resulted in dwarfism (*Figure 3A*), suggesting that the functions of PUF2 and canonical RAB5 lie in the same trafficking pathway.

We previously demonstrated that *ara6-1* and *ara7/rha1* mutations exerted opposing effects on *syp22-1*, a mutation in a SNARE protein mediating membrane fusion at the vacuole, and *vps9a-2*; *ara7* and *rha1* exaggerated the deleterious phenotypes of *syp22-1* and *vps9a-2*, whereas *ara6-1* remedied these phenotypes (*Ebine et al., 2011*). If PUF2 is a downstream effector mediating ARA6 function, then loss of PUF2 function should exert effects similar to that of *ara6-1* on *syp22-1* and *vps9a-2* mutations. However, contrary to this hypothesis, the *puf2 syp22-1* double mutant exhibited more severe wavy-rosette and late-flowering phenotypes than the *syp22-1* mutant (*Figure 3B*), which were restored by introducing the genomic sequence of *PUF2* (*Figure 3—figure supplement 1E and F*). A similar synergistic genetic interaction was also observed between *puf2* and *vps9a-2*; the hemizygous *puf2* mutation aggravated the growth phenotypes of *vps9a-2*, resulting in seedling lethality (*Figure 3C*). Conversely, the *puf2* mutant harboring the hemizygous *vps9a-2* mutation (*puf2 vps9a-2+/−*) exhibited no discernable abnormality, enabling segregation analysis. The observation of developing seeds in seedpods of the *puf2 vps9a-2+/−* mutant revealed that approximately one-fourth of the seeds exhibited a brownish and shriveled appearance (normal:abnormal = 76:25, n = 101 seeds, *p*<0.001, chi-square test, $\chi^2$ = 67.7) (*Figure 3D*), indicating that the double mutation resulted in halted embryogenesis. We subsequently observed embryos collected from *puf2 vps9a+/−* seedpods. Although we were unable to distinguish double mutants from their normal siblings at the globular stage, double-mutant embryos exhibited severe developmental retardation at later developmental stages (*Figure 3E*). The embryonic lethality of the *puf2 vps9a-2* mutant was restored after introducing a genomic fragment containing *PUF2* (*Figure 3—figure supplement 1G and H*).

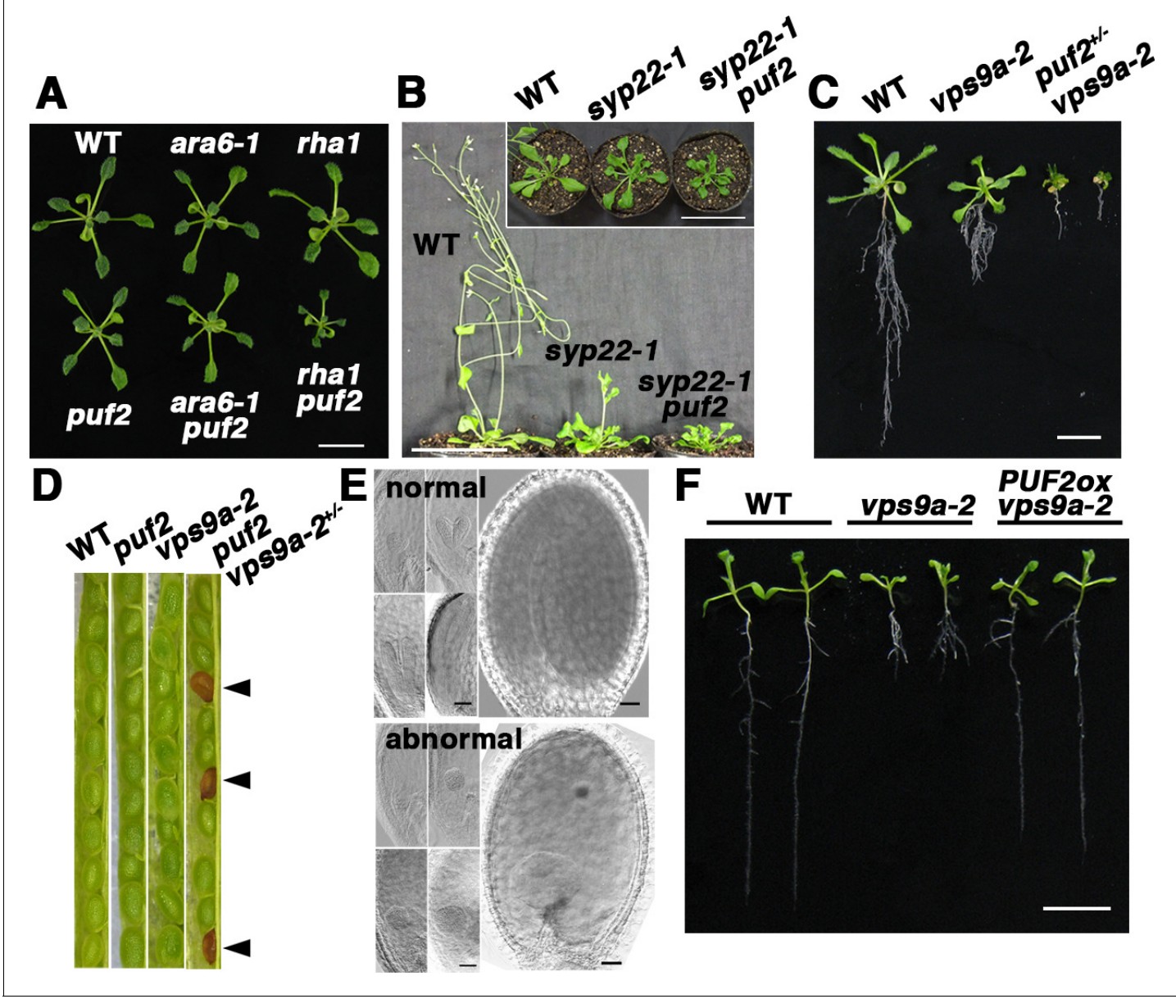

**Figure 3.** Genetic interaction between *PUF2* and endocytic components. (**A**) Genetic interactions between *PUF2* and *ARA6* or *RHA1*. Fourteen-day-old plants are shown. Bar = 1 cm. (**B**) Genetic interaction between *PUF2* and *SYP22*. The *puf2* mutation aggravated the phenotype of the *syp22-1* mutant. The top view of the plants is shown in the inset. Thirty-five-day-old plants are shown. Bars = 5 cm. (**C**) Genetic interaction between *PUF2* and *VPS9a*. Hemizygous mutation of *puf2* (*puf2+/−*) aggravated the *vps9a-2* phenotype. Sixteen-day-old seedlings are shown. Bar = 1 cm. (**D**) Seed pods collected from wild-type, *puf2*, *vps9a-2* and *puf2 vps9a-2+/−* plants. Approximately a quarter of the seeds collected from the *puf2 vps9a-2+/−* plants exhibited a brown and deflated appearance (arrowheads). (**E**) Embryogenesis of the *puf2 vps9a-2* double mutant was delayed and terminated at the globular stage. Corresponding images in the upper and lower sets represent normal and abnormal embryos collected from the same seedpods at different developmental stages. Bars = 50 μm. (**F**) Overexpression of *PUF2* rescued defective primary root elongation of the *vps9a-2* mutant. Ten-day-old seedlings are presented. Bar = 1 cm.

DOI: https://doi.org/10.7554/eLife.34064.008

The following source data and figure supplements are available for figure 3:

**Figure supplement 1.** Related to *Figure 3*.
DOI: https://doi.org/10.7554/eLife.34064.009

**Figure supplement 1—source data 1.** Quantification of primary root length.
DOI: https://doi.org/10.7554/eLife.34064.010

Similar complementation was also observed for the genomic *PUF2* fragment, in which cDNA for GFP was inserted after the start codon or in front of the stop codon, indicating the functionality of GFP-tagged PUF2. An intimate genetic interaction between *PUF2* and *VPS9a* was also observed in this overexpression experiment: overexpressed *PUF2* partially rescued the defective growth of *vps9a-2* (*Figure 3F*, *Figure 3—figure supplement 1I and J*). As the deleterious phenotype of *vps9a-2* reflects the defective activation of canonical RAB5s rather than ARA6 (*Goh et al., 2007*), this genetic evidence indicates that PUF2 acts in the same trafficking event as canonical RAB5, although PUF2 preferentially interacts with GTP-bound ARA6.

## Impact of the *puf2* mutation on vacuolar transport

Canonical RAB5 acts during the transportation of soluble cargos such as 12S globulin and GFP-CT24 to protein storage vacuoles in *Arabidopsis* seeds, a process that also involves SYP22 (*Ebine et al., 2011*; *Ebine et al., 2014*; *Ebine et al., 2008*). If PUF2 indeed acts in the same trafficking pathway as canonical RAB5 and SYP22, then mutations in *PUF2* would synergistically affect the transport of vacuolar cargos in *syp22-1*. We verified this hypothesis by examining the accumulation of storage protein precursors in a *puf2 syp22-1* double mutant. Although processing of storage proteins was not markedly affected in the *puf2* mutant (*Figure 4A*), precursor proteins with larger molecular masses were observed in the *puf2 syp22-1* double mutant (*Figure 4A*, arrowhead). Immunoblot analysis using an anti-12S globulin antibody also revealed the accumulation of 12S globulin precursors in *puf2 syp22-1* and *puf2+/− vps9a-2*, as observed in *rha1 syp22-1* and *vps9a-2* mutants (*Figure 4B* and *Figure 4—figure supplement 1A*). The synergistic interaction between *puf2* and *syp22* or *vps9a-2* mutations was also evident in the mis-secreted GFP-CT24 observed in the seeds of *puf2 syp22-1* and *puf2+/− vps9a-2* plants (*Figure 4C*, *Figure 4—figure supplement 1B*).

Synergistic enhancement of the trafficking defect in *vps9a-2* by *puf2* was also observed in vegetative tissue. It is possible to monitor the constitutive transport of PIN2-GFP from the plasma membrane to the vacuole by the accumulation of GFP fluorescence in the vacuole of root epidermal cells of dark-grown wild-type and *puf2* mutant plants (*Kleine-Vehn et al., 2008*; *Tamura et al., 2003*; *Figure 4D*). Weaker fluorescence was observed in *vps9a-2* plants than in wild-type and *puf2* plants, indicating partial impairment of the endocytic pathway as previously reported (*Inoue et al., 2013*). This trafficking defect was substantially enhanced by hemizygous *puf2* mutation; fluorescence in the vacuole was barely detected, and PIN2-GFP accumulated at punctate or ring-shaped compartments, which likely represent endosomal compartments in the *puf2+/− vps9a-2* mutant (*Figure 4D*, *Figure 4—figure supplement 1C*). These results confirmed that PUF2 acts in the vacuolar/endocytic trafficking pathway with canonical RAB5 and SYP22.

## PUF2 also interacts with VPS9a

The observation that PUF2 acts in the same trafficking pathway as canonical RAB5, even though this protein was isolated as an ARA6 effector, prompted us to determine whether PUF2 integrates the functions of plant-unique and canonical RAB5 groups at endosomes. Therefore, we examined the interactions between PUF2 and other RAB5-related molecules and observed that PUF2 also interacted with VPS9a in a yeast two-hybrid assay (*Figure 5A*). Deletion analysis revealed that this interaction was mediated by the N-terminal coiled-coil region of PUF2 and was distinct from the interaction with GTP-bound ARA6 mediated by the C-terminal coiled-coil region (*Figure 1D*, *Figure 5A*, *Figure 5—figure supplement 1*). The *in planta* interaction between PUF2 and VPS9a was also confirmed by performing co-immunoprecipitation analysis (*Figure 5B*). This interaction was direct: bacterially expressed and purified GST-tagged PUF2[37-127], which contained the N-terminal coiled-coil region, pulled down purified HA-tagged VPS9a, while PUF2[461-639], which contained the C-terminal coiled-coil region, did not (*Figure 5C*). Furthermore, VPS9a and PUF2 colocalized at a subpopulation of endosomes bearing ARA6 (*Figure 5D*, arrowheads).

To verify the significance of the interaction between PUF2 and VPS9a in vacuolar trafficking, we tested the effect of overexpression of the N-terminal coiled-coil region of PUF2 on vacuolar transport of sporamin. When expressed in Arabidopsis suspension cultured cells, sporamin tagged with Venus predominantly accumulated in vacuoles, which was inhibited by co-expression of dominant-negative ARA7 (ARA7[S24N]) (*Figure 5—figure supplement 1B and D*). The N-terminal coiled-coil of PUF2 exerted a similar effect to ARA7[S24N] (*Figure 5—figure supplement 1B and D*). This effect

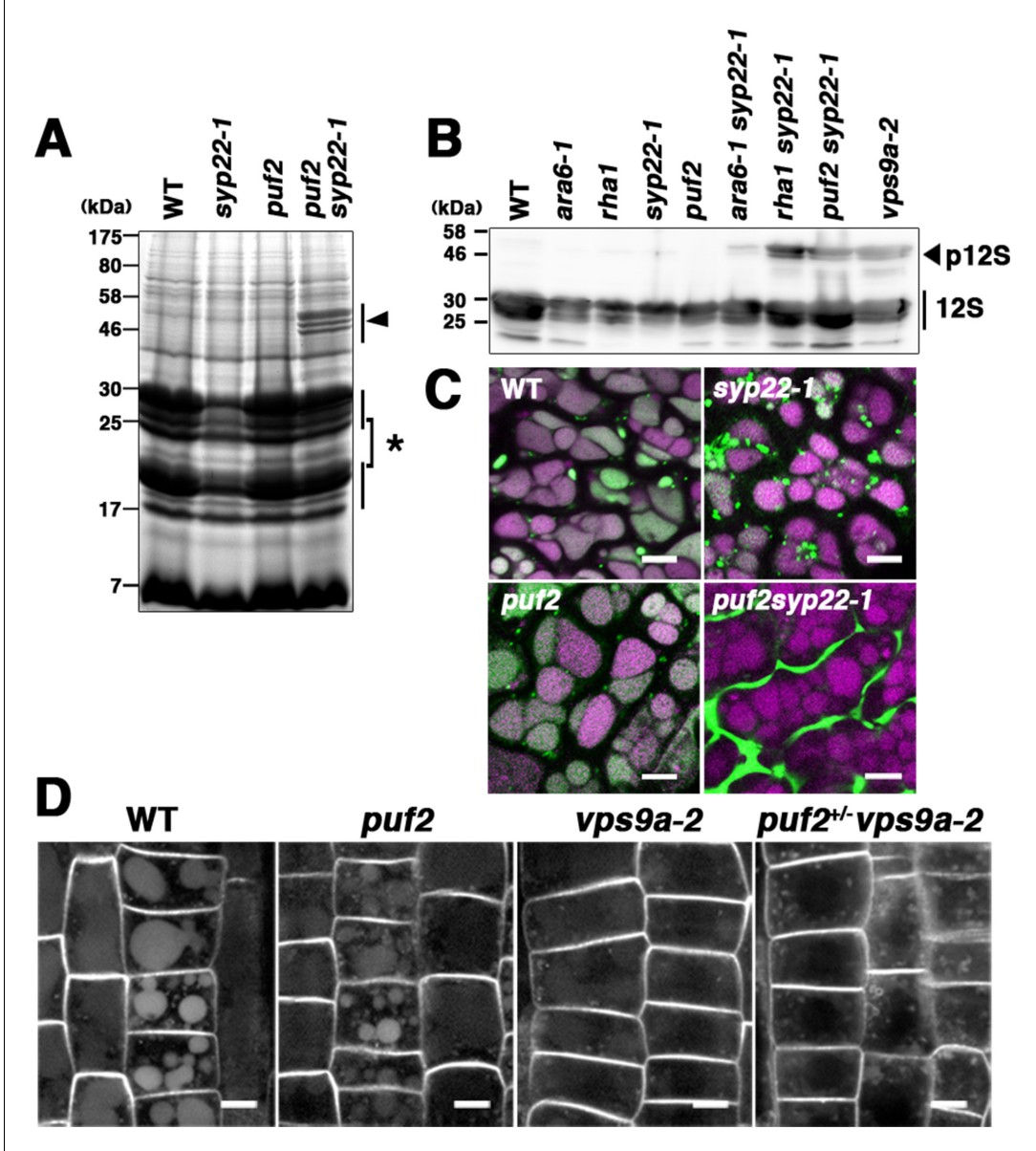

**Figure 4.** PUF2 is involved in vacuolar trafficking of cargo proteins. (A) CBB staining of total seed proteins. Unprocessed precursors of storage proteins with large molecular masses were detected in *puf2 syp22-1* seeds (arrowhead). The asterisk indicates bands for processed storage proteins. (B) Immunoblotting detection of 12S globulin in seed proteins. Precursors of 12S globulin (p12S, arrowhead) accumulated in *puf2 syp22-1* seeds. (C) Distribution of SP-GFP-CT24 in wild-type, *syp22-1*, *puf2*, and *puf2 syp22-1* embryonic cells. SP-GFP-CT24 (green) was mis-secreted into the extracellular space in the *puf2 syp22-1* double mutant. Autofluorescence from protein storage vacuoles is presented in magenta. Bars = 5 μm. (D) Distribution of PIN2-GFP in wild-type, *puf2*, *vps9a-2* and *puf2+/– vps9a-2* root cells. Plants were grown vertically under dark conditions for 48 hr prior to observation. Bars = 5 μm.

DOI: https://doi.org/10.7554/eLife.34064.011

The following source data and figure supplements are available for figure 4:

**Figure supplement 1.** Related to *Figure 4*.
DOI: https://doi.org/10.7554/eLife.34064.012

**Figure supplement 1—source data 1.** Quantification of p12S globulin signal.
DOI: https://doi.org/10.7554/eLife.34064.013

**Figure supplement 1—source data 2.** Quantification of vacuolar PIN2 signal.
DOI: https://doi.org/10.7554/eLife.34064.014

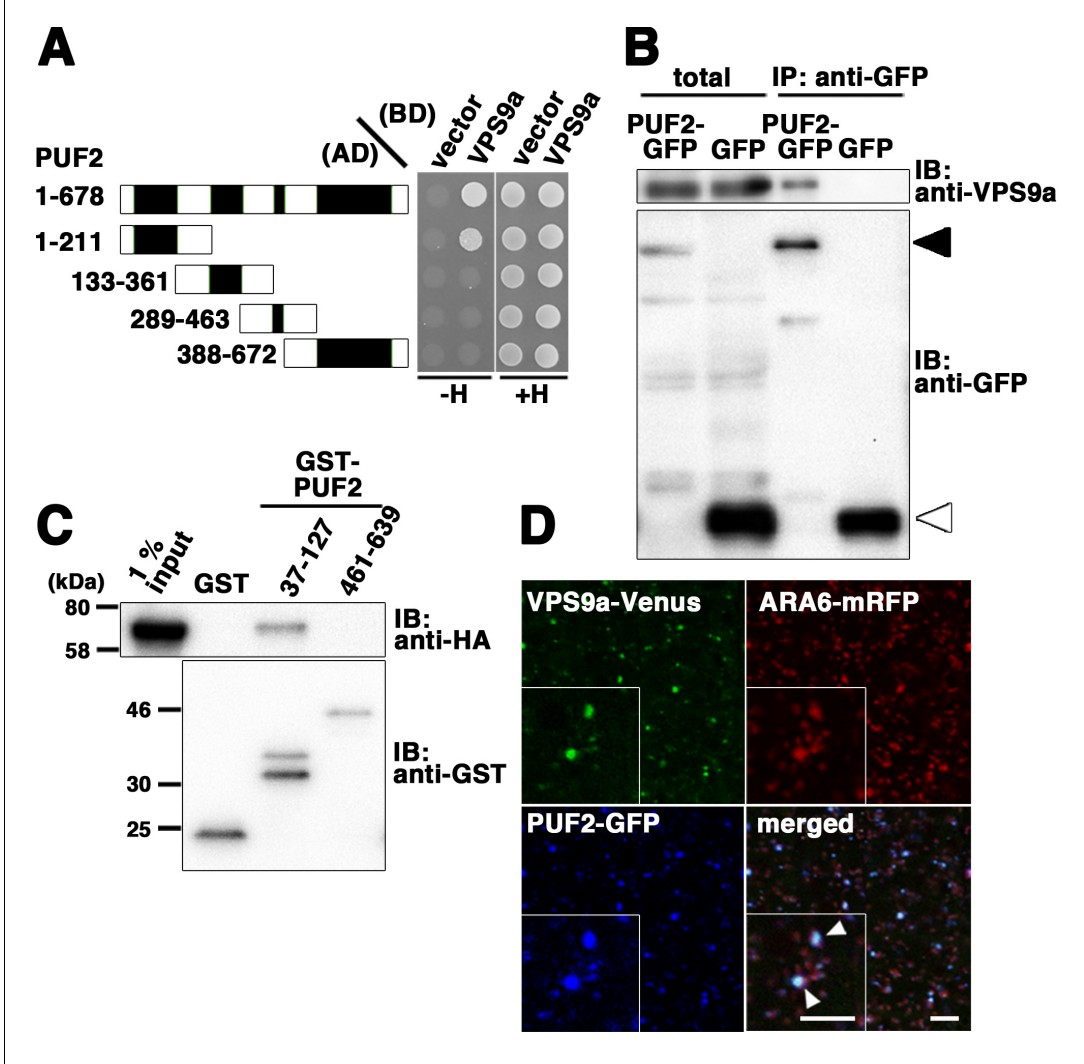

**Figure 5.** Interaction between PUF2 and the RAB5 activating factor VPS9a. (**A**) Yeast two-hybrid interaction between PUF2 and VPS9a. PUF2 containing each coiled-coil region was expressed as a fusion protein with the AD, and VPS9a was expressed as a fusion with the BD in the yeast strain AH109. Interactions were tested using the *HIS3* reporter gene. Black boxes indicate coiled-coil regions. (**B**) Plant lysates prepared from transgenic plants expressing PUF2-GFP (black arrowhead) or free GFP (white arrowhead) were subjected to immunoprecipitation with an anti-GFP antibody, followed by immunoblotting using the indicated antibodies. (**C**) The interaction between PUF2 and VPS9a was detected using an *in vitro* binding assay. Yeast lysate containing VPS9a-HA was subjected to a pull-down assay with GST-fused truncated PUF2 protein. VPS9a-HA interacted with PUF2 37–127 aa containing the N-terminal coiled-coil region but not with PUF2 461–639 aa containing the C-terminal coiled-coil region. VPS9a-HA did not interact with GST. Loading was performed with 1% input. (**D**) Localization of VPS9a-Venus (green), ARA6-mRFP (red), and PUF2-GFP (blue) expressed in the same plant. VPS9a and PUF2 colocalized on a subpopulation of ARA6-positive endosomes. Bars = 5 µm.

DOI: https://doi.org/10.7554/eLife.34064.015

The following source data and figure supplements are available for figure 5:

**Figure supplement 1.** Related to *Figure 5*.

DOI: https://doi.org/10.7554/eLife.34064.016

**Figure supplement 1—source data 1.** Quantification of vacuolar Sporamin-Venus signal.

DOI: https://doi.org/10.7554/eLife.34064.017

should be attributed to titration of VPS9a by the PUF2 N-terminus, because the deleterious effect was suppressed by co-expressing VPS9a-tagRFP, but not by tagRFP alone (*Figure 5—figure supplement 1C and D*). Co-expression of GFP, GFP-tagged full-length PUF2, and the C-terminal coiled-coil region of PUF2 where PUF2 interacts with ARA6 did not exhibit an inhibitory effect on vacuolar

trafficking of sporamin. These results indicated functional significance of the interaction of PUF2 with VPS9a at the N-terminal region in vacuolar transport in Arabidopsis.

## PUF2 recruits VPS9a to endosomes

To investigate the effects of the *puf2* mutation on VPS9a localization, we expressed VPS9a-GFP in the null alleles of the *vps9a* mutant *vps9a-1* (*Goh et al., 2007*) and the *puf2 vps9a-1* double mutant under regulation of the *VPS9a* promoter, which complemented VPS9a function by comparable expression levels of VPS9a-GFP (*Figure 6—figure supplement 1A*). In root epidermal cells of *puf2 vps9a-1*, the endosomal population of VPS9a-GFP was significantly reduced compared to that in *vps9a-1* cells [14.3 ± 1.9 puncta per cell slice in *vps9a-1* (mean ±SD, n = 6 independent images, each of which contained five cells) versus 5.4 ± 2.1 puncta in *puf2 vps9a-1* (mean ±SD, n = 6 independent images, each of which contained five cells), $p<0.01$, Student's t-test] (*Figure 6A*). The remaining VPS9a-GFP-positive dots in *puf2 vps9a-1* plants were sensitive to BFA treatment; however, endosome dilation was rarely observed after Wm treatment (*Figure 6A*, middle and right panels),

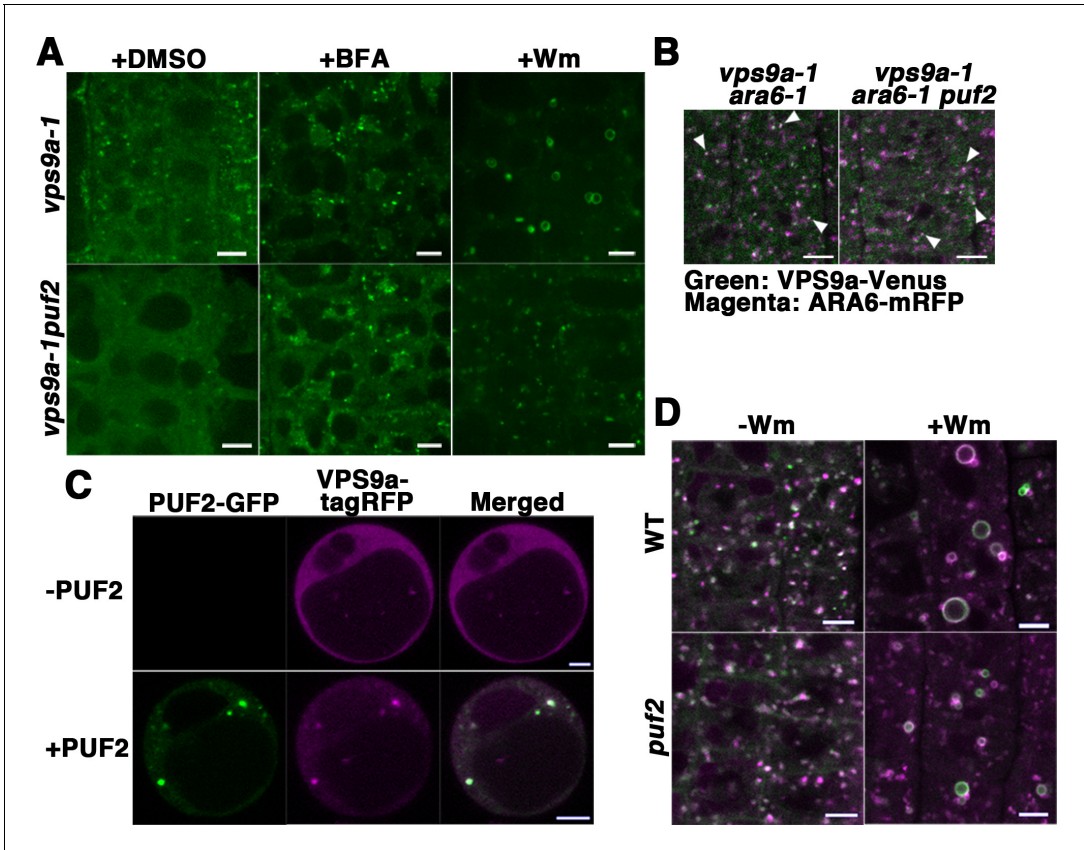

**Figure 6.** PUF2 is required for the efficient recruitment of VPS9a onto endosomes as well as endosomal fusion. (**A**) Localization of VPS9a-GFP in the root epidermal cells of *vps9a-1* and *vps9a-1 puf2* plants treated with DMSO, BFA, or Wm. Bars = 5 μm. (**B**) Localization of VPS9a-Venus (green) and ARA6-mRFP (magenta) in *vps9a-1 ara6-1* and *vps9a-1 ara6-1 puf2* cells. Arrowheads indicate endosomes bearing both VPS9a-GFP and ARA6-mRFP. Bars = 5 μm. (**C**) Overexpression of PUF2-GFP recruits VPS9a onto punctate compartments in protoplasts. VPS9a-tagRFP (magenta) was localized to punctate structures following the overexpression of PUF2-GFP (green) and dispersed throughout the cytosol in the absence of PUF2-GFP. Bars = 5 μm. (**D**) Localization of ARA6-GFP (green) and mRFP-ARA7 (magenta) in the root epidermal cells of the *puf2* mutant treated with DMSO (−Wm) or Wm (+Wm). Bars = 5 μm.

DOI: https://doi.org/10.7554/eLife.34064.018

The following source data and figure supplements are available for figure 6:

**Figure supplement 1.** Related to *Figure 6*.
DOI: https://doi.org/10.7554/eLife.34064.019

**Figure supplement 1—source data 1.** Wortmannin ring size.
DOI: https://doi.org/10.7554/eLife.34064.020

although VPS9a-positive puncta retained their endosomal identity as demonstrated by ARA6 localization (*Figure 6B*). We did not observe a noticeable difference in VPS9a-GFP fluorescence in lateral root cap cells between the *vps9a-1* and *puf2 vps9a-1* mutants (*Figure 6—figure supplement 1B*), likely reflecting distinct PUF2 requirements in different tissues.

According to the results described above, PUF2 promotes the recruitment of VPS9a onto the endosome. This was further confirmed using a transient expression system in protoplasts, in which fluorescently tagged VPS9a (VPS9a-tagRFP) failed to localize on endosomes with dispersed distribution in the cytosol (*Figure 6C*, upper panels). When PUF2-GFP was coexpressed with VPS9a-tagRFP, VPS9a was recruited to PUF2-positive compartments (*Figure 6C*, lower panels), whereas coexpression of another endosomal protein, GFP-VAMP727, did not affect the cytosolic localization of VPS9a (*Figure 6—figure supplement 1C*). Thus, PUF2 facilitates the localization of VPS9a onto endosomes.

## PUF2 is required for endosomal fusion induced by Wm

Reduced endosomal localization of VPS9a in *puf2* may lead to inefficient activation of RAB5 on the endosomal membrane. This hypothesis was verified by examining Wm-induced endosomal fusion (*Wang et al., 2009*) in the *puf2* mutant. Although the distribution of ARA6-GFP and mRFP-ARA7 was not markedly affected in DMSO-treated *puf2* mutant cells (*Figure 6D*, left panels), the diameters of the dilated endosomes induced by Wm treatment were significantly reduced by approximately 30% [$1.98 \pm 0.06$ μm in wild-type (n = 83 dilated endosomes) versus $1.60 \pm 0.03$ μm in mutant cells (n = 206 dilated endosomes), mean $\pm$SD, $p<0.01$, Student's t-test] (*Figure 6D*, right panels). A histogram of the diameters of the dilated endosomes also showed a shift in the peak to the smaller population in the *puf2* mutant (*Figure 6—figure supplement 1D*). Thus, recruitment of VPS9a to the endosomal membrane by PUF2 is required for efficient RAB5 activation, which is likely necessary for Wm-mediated endosomal fusion.

## PUF2 integrates the activities of two RAB5 groups in plants

According to the results described above, PUF2 integrates the functions of ARA6 and canonical RAB5 via the common activator VPS9a; therefore, we next undertook further genetic and biochemical tests. We previously reported that loss of function of *ARA6* suppressed the *vps9a-2* mutation, whose mechanism remains unknown (*Ebine et al., 2011*; *Figure 7*). Here, we examined whether the suppression exerted by the *ara6-1* mutation requires *PUF2*. We analyzed genotypes of progenies generated by the self-pollination of *ara6-1 vps9a-2 puf2+/−* plants and observed that the *ara6-1 puf2 vps9a-2* triple mutant was embryonically lethal (*ara6-1 vps9a-2 puf2+/+:ara6-1 vps9a-2 puf2+/−:ara6-1 vps9a-2 puf2−/−* = 59:110:0, n = 169, $p<0.001$, chi-square test, $\chi^2$ = 15.4) (*Figure 7A*). Thus, PUF2 is required for the suppression of *vps9a-2* by *ara6-1*, and the suppression activity of *ara6-1* is exerted through *PUF2*. Consistently, the *puf2+/− vps9a-2* growth defect was partially suppressed by the *ara6-1* mutation (*Figure 7B*).

Together, these results suggest that PUF2 promotes canonical RAB5-mediated endosomal transport by assembling VPS9a and GDP-bound canonical RAB5 on the endosomal membrane, thereby facilitating the activation of canonical RAB5 on the endosome. GTP-bound ARA6 negatively regulates this process by titrating PUF2, as shown by the effects of different concentrations of ARA6[Q93L] on binding between GST-ARA7[S24N] and PUF2. As the amount of ARA6[Q93L] increased, the amount of PUF2 pulled down by GST-ARA7[S24N] decreased ($57.5 \pm 34.1\%$ band intensity, n = 4 independent experiments, $p<0.05$; *Figure 7C and D*, *Figure 7—figure supplement 1A*). ARA6[S47N] and ARA7[Q69L], non-interactive RAB5 partners of PUF2, did not hamper the interaction between PUF2 and ARA7[S24N] (*Figure 7—figure supplement 1B*). Thus, ARA6 in its active state interferes with the assembly of GDP-bound ARA7, PUF2, and likely VPS9a by competitively binding to PUF2 to diminish endosomal transport mediated by canonical RAB5.

## Discussion

The existence of two RAB5 groups with distinct functions, specifically the canonical and plant-unique RAB5 groups, is a unique characteristic of the plant membrane trafficking system. These RAB5 groups share the upstream regulator VPS9a, containing the VPS9 domain, which is also conserved in activating factors for RAB5 in animal systems (*Carney et al., 2006*; *Goh et al., 2007*; *Ishida et al.,*

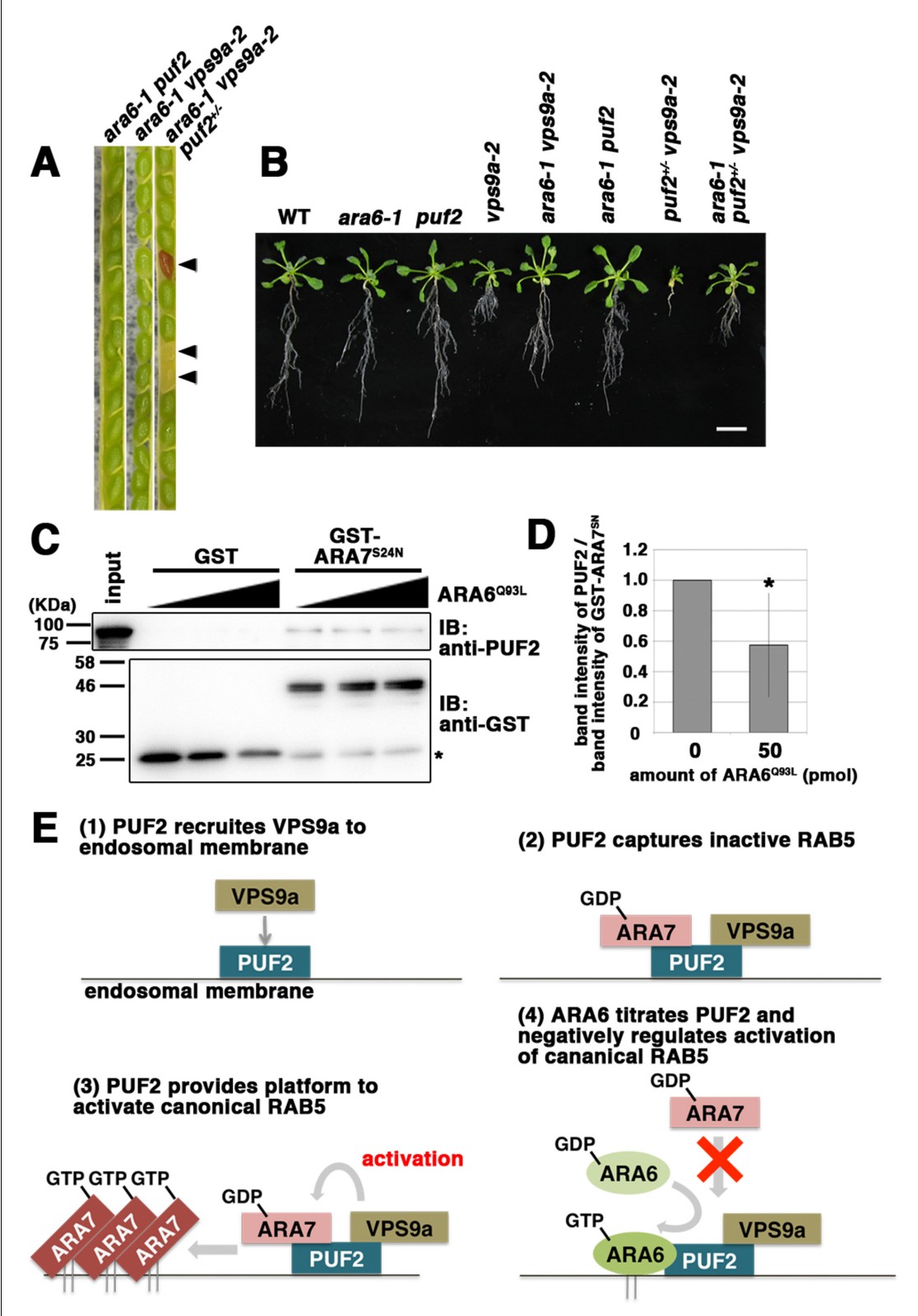

**Figure 7.** PUF2 integrates functions of two plant RAB5 groups. (**A**) Seed pods collected from wild-type, *ara6-1 vps9a-2*, and *ara6-1 vps9a-2 puf2+/−* plants. Arrowheads indicate brown and deflated seeds in a seed pod from the *ara6-1 vps9a-2 puf2+/−* plant. (**B**) Fourteen-day-old seedlings with the indicated genotypes. Bar = 1 cm. (**C**) Interaction between ARA7$^{S24N}$ and PUF2 pre-incubated with ARA6$^{Q93L}$. The assay was performed as described for *Figure 1B*, but PUF2 was pre-incubated with 0 pmol, 25 pmol or 50 pmol ARA6$^{Q93L}$ before mixing with GST-ARA7$^{S24N}$. Loading was performed with

*Figure 7 continued on next page*

*Figure 7 continued*

1.5% input. (D) Quantification of the band intensities of PUF2, calibrated based on the band intensities of GST-ARA7$^{S24N}$. The error bar indicates the mean ±SD. *, $p<0.05$. (F) Proposed model for the integration of two plant RAB5 groups mediated by PUF2. PUF2 recruits VPS9a and the inactive form of canonical RAB5 onto the endosome, leading to the efficient activation of canonical RAB5 and RAB5-dependent trafficking events. GTP-bound active ARA6 negatively regulates this process by competitively binding to the PUF2-VPS9a complex.
DOI: https://doi.org/10.7554/eLife.34064.021
The following source data and figure supplement are available for figure 7:

**Source data 1.** Quantification of PUF2 band intensity.
DOI: https://doi.org/10.7554/eLife.34064.023
**Figure supplement 1.** Related to *Figure 7*.
DOI: https://doi.org/10.7554/eLife.34064.022

*2016*). In animal cells, several VPS9 domain-containing proteins with distinct domain structures regulate Rab5 activity in different endocytic trafficking processes. In plant cells, by contrast, the single activating factor VPS9a coordinates two plant RAB5 groups during vacuolar and endosomal transport, whose molecular mechanisms remain unknown. In the present study, we identified a novel ARA6 effector, PUF2, which acts as a keystone to integrate the functions of two plant RAB5 groups together with VPS9a on the endosomal membrane. PUF2 captures inactive canonical RAB5 and VPS9a on the endosomal membrane, leading to efficient activation of canonical RAB5 and the enhancement of vacuolar transport from endosomes to the vacuole. ARA6, the plant-unique RAB5 in Arabidopsis, diminishes this process by competitively binding to PUF2 in its GTP-bound state against GDP-bound canonical RAB5, thus balancing the distinct canonical RAB5- and ARA6-mediated endosomal transport pathways in plant cells (*Figure 7E*).

No striking homologs of PUF2 have been observed in sequence databases of non-plant systems, suggesting that PUF2 was uniquely acquired by the plant lineage. However, PUF2 shares several characteristics with the animal Rab5 effector protein Rabaptin-5/RABEP1, and both proteins contain four coiled-coil regions. Furthermore, both PUF2 and Rabaptin-5/RABEP1 interact with GTP-bound Rab5 members at one of the coiled-coil regions and also interact with their activating factors at another coiled-coil region (*Horiuchi et al., 1997*). Rabaptin-5 directly binds to the active form of another Rab GTPase, Rab4, at a distinct coiled-coil region from Rab5, suggesting coordination of the two endosomal Rab GTPase functions by this molecule. Although these data suggest functional similarities between PUF2 and Rabaptin-5, PUF2 appears to preferentially interact with inactive canonical RAB5, and GTP-bound active ARA6 represses the action of canonical RAB5 via competitive binding to PUF2. Furthermore, PUF2 interacts with ARA6 and VPS9a at its C-terminal and N-terminal coiled-coil regions, respectively, whereas Rabaptin-5 interacts with Rab5 and Rabex-5 at its fourth and third coiled-coil regions, respectively (*Mattera et al., 2006*; *Vitale et al., 1998*). Rabaptin-5 promotes Rab5 activation by alleviating the autoinhibition of Rabex-5 (*Delprato and Lambright, 2007*; *Lippé et al., 2001*), resulting in additional Rab5 activation to form an active RAB5 domain on the endosomal membrane. Thus, the mechanisms utilized by PUF2 and Rabaptin-5 are completely different. The downregulation of a Rab GTPase by another Rab GTPase via competitive binding with an effector and a common GEF has not previously been reported. These findings demonstrate that plants have developed specific mechanisms to regulate RAB5 GTPases by acquiring two plant-unique machinery components, ARA6 and PUF2, to participate in the underlying mechanism of endocytic pathway regulation in plants.

Although efficient recruitment of VPS9a to the endosome requires PUF2, it is not yet clear how PUF2 is targeted to the endosomal membrane. In the case of Rabaptin-5, Rabaptin-5 interacts with several proteins in addition to Rab5, Rab4, and Rabex-5, including Rabphillin-3 (a Rab3 effector: *Ohya et al., 1998*), γ-adaptin, and GGA (Golgi-localizing, γ-adaptin ear homology domain, ARF-binding proteins) (*Mattera et al., 2003*). The similarity between PUF2 and Rabaptin-5 at the amino acid sequence level is low, and their modes of actions are divergent, but PUF2 may also have other interacting partners because proteins with multiple coiled-coil regions frequently interact with multiple proteins. In the future, it would be interesting to isolate molecules responsible for the upstream regulation of plant RAB5 groups by recruiting PUF2 onto the endosomal membrane. Further screening and characterization of other binding partners of PUF2 is needed. We recently observed an effector of canonical RAB5 in Arabidopsis, EREX, which appears to act only during embryogenesis

and early developmental stages (*Sakurai et al., 2016*), suggesting canonical RAB5 utilizes several distinct effector proteins at different developmental stages. ARA6 may also exert its function by interacting with multiple effector proteins, and the identification and analysis of other ARA6 effectors will allow us to understand the precise molecular mechanisms underlying ARA6 functioning. Such approaches will reveal how plants have pioneered plant-specific endosomal trafficking pathways and provide valuable information regarding the general mechanisms involved in membrane trafficking system diversification during the evolution of eukaryotic organisms.

# Materials and methods

## Key resources table

| Reagent type (species) or resource | Designation | Source or reference | Identifiers | Additional information |
|---|---|---|---|---|
| Gene (*Arabidopsis thaliana*) | PUF2 | NA | TAIR:At1g24560 | |
| Gene (*A. thaliana*) | ARA6/RABF1 | NA | TAIR:At3g54840 | |
| Gene (*A. thaliana*) | RHA1/RABF2a | NA | TAIR:At5g45130 | |
| Gene (*A. thaliana*) | ARA7/RABF2b | NA | TAIR:At4g19640 | |
| Gene (*A. thaliana*) | SYP43 | NA | TAIR:At3g05710 | |
| Gene (*A. thaliana*) | VPS9a | NA | TAIR:At3g19770 | |
| Gene (*A. thaliana*) | SYP22 | NA | TAIR:At5g46860 | |
| Gene (*A. thaliana*) | VAMP727 | NA | TAIR:At3g54300 | |
| Cell line (*A.thaliana*) | Deep | PMID:9681013 | | |
| Genetic reagent (*A.thaliana*) | *puf2* | This paper | TAIR:SAIL_24_C10 | |
| Genetic reagent (*A.thaliana*) | *ara6-1* | PMID:21666683 | TAIR:SAIL_880_C07 | |
| Genetic reagent (*A.thaliana*) | *rha1* | PMID:17468262 | TAIR:SAIL_596_A03 | |
| Genetic reagent (*A.thaliana*) | *syp22-1* | PMID:18984676 | TAIR:SALK_060946 | |
| Genetic reagent (*A.thaliana*) | *vps9a-2* | PMID:18055610 | TAIR:GABI_557C02 | |
| Strain, strain background (*Saccharomyces cerevisiae*) | AH109 | PMID:8978031 | | |
| Strain, strain back ground (*S. cerevisiae*) | YPH414 | PMID:18055610 | | |
| Strain, strain background (*Agrobacterium tumefaciens*) | GV3101::pMP90 | other | | widely distributed |
| Strain, strain background (*Escherichia coli*) | DH5α | PMID:6345791 | | |
| Strain, strain background (*E. coli*) | Rosetta-gami DE3 | Merck | Merck:71351-3CN | |
| Antibody | anti-PUF2; anti-MBP-PUF2 | This paper | | (1:200) |
| Antibody | anti-GFP | This paper | | (1:1,000) |
| Antibody | anti-12S globulin | PMID:14657332 | | (1:10,000) |
| Antibody | anti-HA (rabbit polyclonal) | Themo Fisher Scientific | Themo Fisher Scientific :71–5500; RRID:AB_2533988 | (1:500) |

*Continued on next page*

*Continued*

| Reagent type (species) or resource | Designation | Source or reference | Identifiers | Additional information |
|---|---|---|---|---|
| Antibody | anti-GST (rabbit polyclonal) | Santa Cruz Biotechnology | Santa Cruz Biotechnology :sc-459;RRID:AB_631586 | (1:1,000) |
| Antibody | anti-H3B (rabbit polyclonal) | Merck | Merck: 07–473; RRID:AB_1977252 | (1:1,000) |
| Antibody | anti-ARA6 (rabbit polyclonal) | PMID:17468262 | | (1:200) |
| Antibody | anti-ARA7 (rabbit polyclonal) | PMID:17468262 | | (1:200) |
| Antibody | anti-RHA1 (rabbit polyclonal) | PMID:21666683 | | (1:200) |
| Antibody | anti-VPS9a (rabbit polyclonal) | PMID:18055610 | | (1:1,000) |
| Recombinant DNA reagent | pHGW (vector) | PMID:11992820 | | |
| Recombinant DNA reagent | pGWB (vector) | PMID:17697981 | | |
| Recombinant DNA reagent | pGEX6P-1 (vector) | GE Healthcare | GE Healthcare:28954648 | |
| Recombinant DNA reagent | pTS911 (vector) | PMID:26493488 | | |
| Commercial assay or kit | GAL4 Two-Hybrid Phagemid Vector Kit | Agilent Technologies | Agilent Technologies:211351 | |
| Commercial assay or kit | Protein Fusion and Purification System | NewEngland BioLabs | NewEngland BioLabs:800 | |
| Commercial assay or kit | uMACS GFP isolation kit | Miltenyi Biotec | Miltenyi Biotec:130-091-288 | |
| Chemical compound, drug | DSP | Themo Fisher Scientific | Themo Fisher Scientific:22585 | |
| Software, algorithm | Fiji | PMID:22743772 | RRID:SCR_002285 | |
| Software, algorithm | SMART | PMID:25300481 | RRID:SCR_005026 | |
| Other | Wortmannin | Sigma-Aldrich | Sigma-Aldrich:W1628-1MG | |
| Other | Brefeldin A | Sigma-Aldrich | Sigma-Aldrich:B7651-5MG | |
| Other | FM4-64 | Themo Fisher Scientific | Themo Fisher Scientific:T13320 | |

## Yeast two-hybrid interaction assays

Individual interaction assays were performed using GAL4 Two-Hybrid Phagemid Vector Kits (Agilent Technologies, Santa Clara, California, USA) and the AH109 strain (Takara Clontech, Kusatsu, Siga, Japan). The colonies were cultured in selective medium without leucine and tryptophan (designated as '+H'), diluted to OD600 = 0.5, and spotted onto plates containing media without leucine, tryptophan, and histidine (designated as '–H') as well as +H plates. More than three independent colonies were tested for each interaction.

## Plant materials and plasmids

The Arabidopsis *puf2* (SAIL_24_C10) mutant was obtained from ABRC (*Alonso et al., 2003*) and backcrossed more than three times with wild-type Arabidopsis (Col-0) prior to use in subsequent experiments. *ara6-1*, *rha1*, *syp22-1*, and *vps9a-2* were obtained from our lab stock. A 5.9-kbp *PUF2* genomic fragment, including 2.0-kbp 5' and 0.9-kbp 3' flanking sequences, was subcloned into the pHGW vector (*Karimi et al., 2002*) and used for the complementation assay and overexpression analysis. The translational fusions of PUF2 with GFP and ARA6 with Venus were prepared by adding fluorescence tags to the full-length proteins (*Tian et al., 2004*). The cDNA for GFP was fused in front of the initiation or stop codons in the *PUF2* genomic sequence as described above to produce GFP-

PUF2 and PUF2-GFP, respectively. The cDNA for Venus was inserted in front of the stop codon of the 5.7 kb genomic fragment of *ARA6* (At3g54840), which included a 2.7 kb promoter, exons, introns, and 3' flanking region, to generate ARA6-Venus. The chimeric fragments were subcloned into the pGWB1 vector, a kind gift from Dr T. Nakagawa (Shimane University, Japan). Transformation of Arabidopsis plants was performed by floral dipping (*Clough and Bent, 1998*), using the *Agrobacterium tumefaciens* strain GV3101::pMP90. Transgenic lines expressing ARA6-mRFP, ARA6$^{Q93L}$-mRFP, ARA6$^{S47N}$-mRFP, mRFP-ARA7, mRFP-SYP43, ST-mRFP, VPS9a-GFP and/or VPS9a-Venus were generated as described previously (*Ebine et al., 2011*; *Ebine et al., 2008*; *Inada et al., 2016*; *Sunada et al., 2016*; *Uemura et al., 2012*). Transgenic plants expressing SP-GFP-CT24 and PIN2-GFP were kind gifts from Dr S. Utsumi (Kyoto University, Japan) and Dr J. Friml (IST, Austria), respectively, and were crossed with *puf2 vps9a-2+/−* and *puf2 syp22-1*. The plants were grown on Murashige and Skoog medium [MS medium: 1 × MS salt (Wako), 2% sucrose, 1 × Gamborg's vitamin solution (Sigma), adjusted to pH 6.3] at 23°C under constant light.

## Microscopy

Transgenic plants expressing GFP-, Venus-, and/or mRFP-tagged proteins were mounted in 1/2 × MS liquid medium and observed under an LSM710, LSM780 (Carl Zeiss, Oberkochen, Germany), or a microscope (model BX51; Olympus, Shinjuku, Tokyo, Japan) equipped with a confocal scanner unit (model ORCA-AG; Yokogawa Electric, Musashino, Tokyo, Japan). At least three different seedlings from three independent transgenic lines were observed for microscopy. For drug treatments, 5 day-old seedlings were soaked in 1/2 × MS liquid medium containing 3.3 µM wortmannin (Sigma-Aldrich, St. Louis, Missouri, USA) or 50 µM brefeldin A (Sigma-Aldrich) for two hours or one hour, respectively. For FM4-64 labeling, 5-day-old seedlings were treated with 4 µM FM4-64 (Thermo Fisher Scientific, Waltham, MA) at 23°C for 30 min. The Feret's diameters of Wm rings were measured using ImageJ software (National Institutes of Health, Maryland, Washington, DC). Transient expression of GFP- and/or tagRFP-tagged proteins in Arabidopsis suspension cultured cells was performed as described previously (*Ueda et al., 2001*; *Ueda et al., 2004*). The Arabidopsis Col-0 suspension cultured line (Deep) was described in *Mathur et al. (1998)*. The amounts of plasmids used for transformation were as follows: 10 µg for *GFP*, PUF2-*GFP*, PUF2$^{1-211}$-*GFP*, PUF2$^{388-672}$-*GFP*, and *Sporamin-Venus* subcloned in pHTS13, 30 µg for *tagRFP*, *VPS9a-tagRFP*, and *tagRFP-ARA7$^{S24N}$* subcloned in pHTS13, and 1 µg for *GFP-VAMP727* subcloned in pUC18 (*Ueda et al., 2004*; *Uemura et al., 2004*). Ten different cells that were successfully transformed were observed. Whole-mount visualization of embryos was performed as previously described (*Aida et al., 1997*). To monitor the vacuolar targeting of PIN2-GFP, transgenic plants were initially grown vertically for 4 days under constant light and subsequently incubated in the dark for 48 hr. Vacuolar accumulation of PIN2-GFP or sporamin-Venus was measured using Fiji (*Schindelin et al., 2012*). At least five different seedlings form each line were observed.

## Expression and purification of recombinant proteins

ARA6$^{Q93L}$, ARA6$^{S47N}$, ARA7$^{Q69L}$, ARA7$^{S24N}$, PUF2 (37–127 aa), and PUF2 (461–639 aa) were expressed as glutathione S-transferase (GST)-fusion proteins in *Escherichia coli* strain DH5α and purified according to the manufacturer's instructions (GE Healthcare, Little Chalfont, Buckinghamshire, England). To obtain full-length PUF2 protein, the codon usage of PUF2 cDNA was optimized to that of *E. coli*, and subsequently codon-optimized *PUF2* was subcloned into the pGEX6P-1 vector (GE Healthcare). GST-fusion protein was expressed in *E. coli* strain Rosetta-gami (DE3) (Merck, Darmstadt, Germany), and GST-PUF2 was digested using the PreScission protease (GE Healthcare) on resin. Eluted PUF2 was concentrated using Amicon Ultra-15 Centrifuge Filter Units (Merck).

## *In vitro* pull-down assay

GST-RAB5s (0.2 nmol), pre-bound to glutathione sepharose 4B resin (GE Healthcare), were incubated in buffer A [20 mM Tris-HCl pH, 7.5, 150 mM NaCl, 5 mM MgCl$_2$, and 0.05% Tween-20] containing 50 mM EDTA and 100 µM GTPγS or GDP and then incubated with PUF2 (0.266 nmol) in buffer A for 30 min at room temperature. The beads were washed three times with buffer A containing 10 µM GTPγS or GDP, and bound proteins were subjected to immunoblotting analysis. For an *in vitro* competition assay, the ARA6$^{Q93L}$, ARA6$^{S47N}$, or ARA7$^{Q69L}$ were mixed with PUF2 and

incubated in buffer A containing 10 µM GTPγS (for ARA6$^{Q93L}$ and ARA7$^{Q69L}$) or GDP (for ARA6$^{S47N}$) prior to mixing with GDP-preincubated GST-ARA7$^{S24N}$. HA-tagged VPS9a was expressed in the yeast strain YPH414 (MATa Δpep4:TRP1 ura3 lys2 ade2 trp1 his3 leu2) under the control of the GAL1 promoter. Yeast cells were collapsed by vortexing with glass beads in PBS with protease inhibitor cocktail (GE Healthcare). The collected lysates (475 µg) were mixed with GST or GST-tagged truncate PUF2 (0.2 nmol), which were pre-bound to glutathione-Sepharose 4B resin (GE Healthcare), and incubated for 60 min at 4°C in PBS buffer containing 0.05% Tween-20 and protease inhibitor cocktail (GE Healthcare). The beads were washed three times with the same PBS buffer, and the bound proteins were subjected to immunoblotting analysis. At least three independent experiments were performed.

## Immunoprecipitation

T3 plants expressing PUF2-GFP were grown vertically on MS medium plates for 16 days, and 0.6 g of each sample was collected and ground in 1 ml of extraction buffer [50 mM HEPES-KOH, pH 7.5, 0.4 M sucrose, 5 mM MgCl$_2$, protease inhibitor cocktail (Roche, Basel, Switzerland)] using sea sand. The lysates were centrifuged at ×1000 g to remove debris. For analysis using a chemical cross-linker, 1 mM DSP (Thermo Fisher Scientific) was added to the supernatants and incubated for 30 min at 4°C, followed by quenching using 50 mM Tris-HCl, pH 7.5, and 0.5% CHAPS was added to the lysates to solubilize the membranes. Next, 750 µl of each sample was incubated with 50 µl of anti-GFP micro beads (Miltenyi Biotec, Bergisch Gladbach, Germany) for 30 min at 4°C. The samples were loaded onto microcolumns attached to the magnetic field of a micro-MACS separator (Miltenyi Biotec) and washed four times with extraction buffer supplemented with 0.5% CHAPS. Immunoprecipitates were eluted according to the manufacturer's instructions, followed by immunoblotting. At least three independent experiments were performed.

## Antibodies

MBP-tagged truncated PUF2, which included amino acid residues 133 to 361, was expressed in *E. coli* Rosetta-gami (DE3) (Merck), purified according to the manufacturer's instructions (New England Biolabs, Ipswich, Massachsetts, USA), and subsequently used as an antigen to generate an anti-PUF2 polyclonal antibody. The obtained anti-PUF2 antibody was purified via protein G affinity column chromatography (GE Healthcare). The anti-GFP antibody was raised against GST-tagged GFP, which was expressed in *E. coli* Rosetta (DE3) (Merck) according to the manufacture's instructions (GE Healthcare). The obtained anti-GFP antibody was purified with HiTrap NHS-activated HP columns (GE Healthcare) conjugated with the purified GFP protein. The anti-12S globulin antibody was a kind gift from Dr I. Hara-Nishimura (Kyoto University, Japan). Anti-HA, anti-GST, and anti-H3B antibodies were purchased from Thermo Fisher Scientific, Santa Cruz Biotechnology (Dallas, TX), and Merck, respectively. The following dilution ratios were used for each antibody in the immunoblotting experiments: anti-ARA6 (*Haas et al., 2007*), 1:200; anti-RAB5 (mixture of anti-RHA1 (*Ebine et al., 2011*), 1:1000 and anti-ARA7 (*Haas et al., 2007*), 1:500); anti-GFP, 1:1,000; anti-12S globulin (*Shimada et al., 2003*), 1:10,000; anti-VPS9a (*Goh et al., 2007*), 1:1,000; anti-HA, 1:500; anti-GST, 1:1,000; anti-H3B, 1:1,000; and anti-PUF2, 1:200.

## Accession numbers

The *Arabidopsis* Genome Initiative locus identifiers for the genes utilized in this study are At1g24560 (PUF2), At3g54840 (ARA6/RABF1), At5g45130 (RHA1/RABF2a), At4g19640 (ARA7/RABF2b), At3g05710 (SYP43), At3g19770 (VPS9a) and At5g46860 (VAM3/SYP22).

## Acknowledgements

We thank Dr T Demura (NAIST, Japan), Dr M Yamaguchi (Saitama University, Japan), Dr T Nakagawa (Shimane University, Japan), Dr J Friml (IST, Austria), Dr S Utsumi, Dr I Hara-Nishimura (Kyoto University, Japan), Dr T Uejima, Dr K Ihara, and Dr S Wakatuski (High Energy Accelerator Research Organization, Japan), Dr S Mano (NIBB, Japan), for sharing materials, as well as the SALK Institute, the Max Planck Institute, and the ABRC for providing the Arabidopsis mutants. This work was financially supported by Grants-in-Aid for Scientific Research from the Ministry of Education, Culture, Sports, Science, and Technology of Japan (to AN, 25221103, TU, 24114003 and 24370019, and EI, 15K18527

and 17K15144), JST, PRESTO (to TU, JPMJPR11B2), a Grant-in-Aid for JSPS fellows (EI, 2010649), the Mitsubishi Foundation, Yamada Science Foundation, Kato Memorial Bioscience Foundation, NIBB Collaborative Research Program (16–339, 17–302, 18–302 to EI), and the Building of Consortia for the Development of Human Research in Science and Technology, MEXT, Japan.

## Additional information

### Funding

| Funder | Grant reference number | Author |
|---|---|---|
| Japan Society for the Promotion of Science | 2010649 | Emi Ito |
| Kato Memorial Bioscience Foundation | | Emi Ito |
| NIBB Collaborative Research Program | 16–339, 17–302, 18–302 | Emi Ito |
| Ministry of Education, Culture, Sports, Science and Technology | 15K18527 | Emi Ito |
| Ministry of Education, Culture, Sports, Science and Technology | 17K15144 | Emi Ito |
| Ministry of Education, Culture, Sports, Science, and Technology | 25221103 | Akihiko Nakano |
| Japan Science and Technology Agency | JPMJPR11B2 | Takashi Ueda |
| Mitsubishi Foundation | | Takashi Ueda |
| Yamada Science Foundation | | Takashi Ueda |
| Ministry of Education, Culture, Sports, Science and Technology | 24114003 | Takashi Ueda |
| Ministry of Education, Culture, Sports, Science and Technology | 24370019 | Takashi Ueda |

The funders had no role in study design, data collection and interpretation, or the decision to submit the work for publication.

### Author contributions

Emi Ito, Conceptualization, Funding acquisition, Investigation, Writing—original draft; Kazuo Ebine, Resources, Writing—review and editing; Seung-won Choi, Investigation; Sakura Ichinose, Tomohiro Uemura, Resources; Akihiko Nakano, Supervision, Funding acquisition, Writing—review and editing; Takashi Ueda, Conceptualization, Supervision, Funding acquisition, Writing—original draft, Writing—review and editing

### Author ORCIDs

Emi Ito iD http://orcid.org/0000-0003-0536-5239
Akihiko Nakano iD http://orcid.org/0000-0003-3635-548X
Takashi Ueda iD http://orcid.org/0000-0002-5190-892X

### Decision letter and Author response

Decision letter https://doi.org/10.7554/eLife.34064.026
Author response https://doi.org/10.7554/eLife.34064.027

## Additional files

**Supplementary files**
• Transparent reporting form
DOI: https://doi.org/10.7554/eLife.34064.024

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
