## [Decision Letter]

Thank you for submitting your article "Integration of Two RAB5 Groups during Endosomal Transport in Plants" for consideration by *eLife*. Your article has been favorably evaluated by Christian Hardtke (Senior Editor), a Reviewing Editor, and three reviewers. The reviewers have opted to remain anonymous.

The reviewers have discussed the reviews with one another and the Reviewing Editor has drafted this decision to help you prepare a revised submission.

All reviewers agree that the identification of PUF2 as a novel interactor of canonical and non-canonical RAB5s and its subsequent impact on VPS9a is highly interesting. Moreover, the work is in principle of high quality and novel. However, they also all pinpoint that the titration model is not well enough supported by the presented data.

We would like to invite you to resubmit a revised version, which provides additional support for the titration model and may balance its discussion.

Please, see below the list of prioritised reviewer suggestions:

1) The reviewers would like you to expand on the titration assays with more controls: The increase of the titrating agent should be demonstrated and we would like to see evidence for the GDP/GTP state dependent competition. You could compare ARA6-GDP and ARA6-GTP in their capacity to titrate the PUF2-ARA7-GDP complex. Ideally, you should do the same on a PUF2-ARA7-GTP complex or show that such a complex does not form in your in vitro system.

2) To validate the scaffolding function of Puf2 in planta, the reviewers propose to use an N terminal and C terminal truncations of Puf2. If your model is correct, overexpression of the Puf2 N terminal CC domain fragment should trap Vps9 and lead to defects in vacuolar transport. Similarly, overexpression of Puf2 c terminal fragment should trap the Rabs and inhibit their activation, since Vps9 will not be recruited. We envision that transient assays would be sufficient in the course of this revision.

3) Some data, such as the vacuolar accumulation of PIN2-GFP, requires detailed quantification.

---

## [Author Response]

1) The reviewers would like you to expand on the titration assays with more controls: The increase of the titrating agent should be demonstrated and we would like to see evidence for the GDP/GTP state dependent competition. You could compare ARA6-GDP and ARA6-GTP in their capacity to titrate the PUF2-ARA7-GDP complex. Ideally, you should do the same on a PUF2-ARA7-GTP complex or show that such a complex does not form in your in vitro system.

We very much appreciate the reviewers’ suggestion, and performed further biochemical assays. We successfully detected the dosage-dependent decrease in the interaction between PUF2 and ARA7^S24N^ by adding ARA6^Q93L^. We also performed the biochemical assay using non-interactive partners of PUF2, ARA6^S47N^ and ARA7^Q63L^. In either case, our assay indicated that ARA6^S47N^ and ARA7^Q63L^ did not show significant effect onto the interaction between PUF2 and ARA7^S24N^. These results are mentioned in the text (subsection “PUF2 integrates the activities of two RAB5 groups in plants”, last paragraph) and presented as Figure 7—figure supplement 1 in the revised manuscript.

2) To validate the scaffolding function of Puf2 in planta, the reviewers propose to use an N terminal and C terminal truncations of Puf2. If your model is correct, overexpression of the Puf2 N terminal CC domain fragment should trap Vps9 and lead to defects in vacuolar transport. Similarly, overexpression of Puf2 c terminal fragment should trap the Rabs and inhibit their activation, since Vps9 will not be recruited. We envision that transient assays would be sufficient in the course of this revision.

We thank the reviewers for this suggestion. We performed the transient expression analysis using suspension-cultured cells derived from Arabidopsis root cells to verify the possible effects of N- and C-terminal regions of PUF2 on vacuolar trafficking. As predicted by the reviewers, we detected an inhibitory effect for the N-terminal coiled-coil of PUF2, which binds to VPS9a but not GTP-ARA6, on vacuolar transport of sporamin. This inhibitory effect should be conferred by titration of VPS9a, because co-expression of VPS9a suppressed the impaired vacuolar transport. This result nicely supports our model that PUF2 regulates vacuolar trafficking via the VPS9a function. We also examined the effect of overexpression of the C-terminal coiled-coil, which binds to GTP-bound ARA6 but not VPS9a, and found that it did not exert a negative effect onto vacuolar transport of sporamin, consistent with our previous finding that ARA6 is dispensable for vacuolar trafficking in *A. thaliana* (Ebine et al., 2011). These results are mentioned in the text (subsection “PUF2 also interacts with VPS9a”, last paragraph) and presented in Figure 5—figure supplement 1.

3) Some data, such as the vacuolar accumulation of PIN2-GFP, requires detailed quantification.

According to the comment, we quantified vacuolar accumulation of PIN2 in wild-type, *puf2, vps9a-2*, and *puf2+/- vps9a-2* plants. The result is shown in Figure 4—figure supplement 1C.